# In Vitro Alpha-Glucosidase Inhibitory Activity and the Isolation of Luteolin from the Flower of *Gymnanthemum amygdalinum* (Delile) Sch. Bip ex Walp.

**DOI:** 10.3390/molecules27072132

**Published:** 2022-03-25

**Authors:** Sheppriola Vonia, Rika Hartati, Muhamad Insanu

**Affiliations:** Pharmaceutical Biology Department, School of Pharmacy, Institut Teknologi Bandung, Ganesha 10, Bandung 40132, Indonesia; vonia.she@gmail.com (S.V.); rika@fa.itb.ac.id (R.H.)

**Keywords:** asteraceae, diabetes, *Gymnanthemum amygdalinum*, luteolin, *Vernonia amygdalina*

## Abstract

Diabetes mellitus is a major health issue that has posed a significant challenge over the years. *Gymnanthemum amygdalinum* is a well-known plant that can be potentially used to treat this disease. Therefore, this study aimed to evaluate the inhibitory effect of its root, stem bark, leaves, and flower extracts on alpha-glucosidase using an in vitro inhibition assay to isolate the bioactive compounds and determine their levels in the samples. The air-dried plant parts were extracted by maceration using methanol. The results showed that the flower extract had the greatest inhibitory effect (IC50 47.29 ± 1.12 µg/mL), followed by the leaves, roots, and stem bark. The methanolic flower extract was further fractionated with different solvents, and the ethyl acetate fraction showed the strongest activity (IC50 19.24 ± 0.12 µg/mL). Meanwhile, acarbose was used as a positive control (IC50 73.36 ± 3.05 µg/mL). Characterization based on UV, 1H-, and 13C-NMR established that the ethyl acetate fraction yielded two flavonoid compounds, namely, luteolin and 2-(3,4-dihydroxyphenyl)-5,7-dihydroxy-3-methoxy-4H-chromen-4-on, which had IC50 values of 6.53 ± 0.16 µg/mL and 39.95 ± 1.59 µg/mL, respectively. The luteolin levels in the crude drug, methanolic extract, and ethyl acetate fraction were 3.4 ± 0.2 mg (0.3%), 32.4 ± 0.8 mg (3.2%), and 68.9 ± 3.4 mg (6.9%) per 1 g samples, respectively. These results indicated that the *G. amygdalinum* flower extract exerted potent inhibitory alpha-glucosidase activity.

## 1. Introduction

As a metabolic disease, type 2 diabetes mellitus is reported in more than 90% of diabetes cases worldwide [1] and is characterized by elevated postprandial blood glucose levels [2]. Therefore, one effective strategy to control postprandial hyperglycemia is delaying glucose absorption using alpha-glucosidase inhibitors (AGIs). Examples of these drugs include acarbose, miglitol, and voglibose [3]. Since AGIs cause side effects such as gastrointestinal disorders [4,5], an effective alternative treatment with fewer side effects and proven safety is needed [6]. Consequently, searching for new AGI sources is very important for therapeutic applications. Traditionally, several plants are used as potential alpha-glucosidase inhibitors [7]. One of these is *Gymnanthemum amygdalinum*, also known as *Vernonia amygdalina*, which belongs to the Asteraceae family. It possesses various pharmacological activities. It is for constipation, diarrhea, skin wounds, scabies, ascariasis, tonsillitis, fever, and malaria [8,9,10]. Several studies reported that *G. amygdalinum* was traditionally used for managing diabetes mellitus in Nigeria [11,12]. Some secondary metabolites can be isolated from *G. amygdalinum* leaves, such as vernonioside A1, vernonioside A2, vernonioside B1, vernonioside B2, vernodalin, vernolepin, vernomygdin, vernodalol, vernodalinol, vernoamyoside, luteolin, luteolin 7-O-β-glucoside, and luteolin 7-O-glucuronide [10]. Furthermore, the flavonoid-rich fraction of *G. amygdalinum* leaf extracts show a significant antidiabetic effect [9]. Luteolin is a flavonoid compound that is well known to be an alpha-glucosidase inhibitor. Previous studies show that the number of phenolic hydroxyl groups on the B ring is correlated with inhibitory activity [13]. This property is possibly due to the alpha-glucosidase inhibitory activity found in the leaf extract, but this is yet to be confirmed for the plant’s other parts. Therefore, this study aimed to evaluate the in vitro AGI activity of *G. amygdalinum* roots, stem barks, leaves, and flowers. The active compounds were isolated from the selected parts, and their levels were determined through thin-layer chromatography (TLC) densitometry.

## 2. Results

### 2.1. Phytochemical Screening

The phytochemical screening of methanolic flower extracts detected alkaloid, flavonoid, tannin, quinone, and triterpenoid.

### 2.2. In Vitro Alpha-Glucosidase Inhibitory Activity

The alpha-glucosidase inhibitory activities of the methanolic crude extracts of G. amygdalinum roots, stem bark, leaves, and flowers were determined using pNPG for the substrate (Table 1). Each extract’s test results are shown by percentage inhibition values, while the data for the extract with the most significant inhibitory effect are also presented with IC_50_ values. Based on the results, the flower extract had better activity than acarbose. The flower fractions’ alpha-glucosidase inhibitory activity showed that ethyl acetate had the greatest inhibitory effect, followed by *n*-hexane and the water fraction (Table 2). The flower extract with an IC_50_ value of 47.29 ± 1.12 µg/mL (Table 3) had the greatest inhibitory effect, followed by the leaf, stem bark, and root extracts (Table 1). This value was lower compared to the positive control, acarbose, which recorded 72.99 ± 2.77 µg/mL (Table 3).

Based on its activity, the flower’s crude methanolic extract was further fractionated successively in different polar and non-polar solvents. The ethyl acetate fraction with an IC_50_ value of 19.24 ± 0.12 µg/mL (Table 3) exerted a significant inhibitory effect on alpha-glucosidase. It yielded two flavonoids, with this being followed by *n*-hexane and water (Table 2). Further, the bioactive compounds’ structures were analyzed using UV and NMR spectral analysis, as shown in Figure 1.

Compound **1** was obtained as a yellow crystal, eluted using *n*-hexane/ethyl acetate/formic acid (4:1:0.05), with this being followed by its visualization in a yellow spot form by spraying it with 10% H_2_SO_4_ and citroborate on TLC silica gel F_254_. This compound showed that UV maxima at 269 and 355 nm supported the presence of the flavonoid skeleton. Compound **1**’s 1H-NMR spectrum (DMSO-d_6_) showed a signal at δ 12.98 (1H, s), which served as evidence for the presence of 5-OH, with the chelated hydroxy group causing this. The singlet peak at δ 6.67 (1H, s, H-3) was identified as hydrogen at C-3 in the flavones. Furthermore, three aromatic proton signals were detected at δ 7.42 (1H, d, *J* = 2.35 Hz, H-2′), 6.89 (1H, d, *J* = 8.25 Hz, H-5′), and 7.40 (1H, dd, *J* = 2.35, 7.50 Hz, H-6′), respectively. The other aromatic proton signals were detected at δ 6.19 (1H, d, *J* = 2.15 Hz, H-6) and 6.44 (1H, d, *J* = 2.10 Hz, H-8). The presence of α, β-unsaturated ketone was evident from the appearance of carbonyl carbon signals at δ 181.7 (C-8). Additionally, the 13C-NMR spectrum (DMSO-d_6_) was observed at δ: 93.8 (C-8), 98.8 (C-6), 102.9 (C-3), 103.7 (C-10), 113.4 (C-2′), 116.0 (C-5′), 119.0 (C-6′), 121.5 (C-1′), 145.7 (C-3′), 149.7 (C-4′), 157.3 (C-9), 161.5 (C-5), 163.9 (C-2), and 164.1 (C-7). The presence of 15 carbon resonances was shown by this expression, which strongly agrees with a previous report on 2-(3,4-dihydroxyphenyl)-5,7-dihydroxy-4H-chromen-4-one [(luteolin) (C_15_H_10_O_6_)] [14].

Compound **2** was obtained as a yellow crystal, eluted thrice using *n*-hexane/ethyl acetate/formic acid (35:15:0.5) as an eluent, and visualized as a dark spot under UV at λ 366 nm in TLC silica gel F254. The dark spot changed to yellow after it was sprayed with citroborate. These results show that both compounds have a flavonoid structure but do not have a free OH group on C-3. Further, the absorption maxima at 269 and 355 nm confirmed the compound’s flavonol nature. The proton NMR spectrum analysis detected three aromatic proton signals at δ 7.63 (1H, d, *J* = 2.20 Hz, H-2′), 6.91 (1H, d, *J* = 8.45 Hz, H-5′), and 7.54 (1H, dd, *J* = 2.20; 8.50 Hz, H-6′), with these signals being typical of flavonoids with 3′,4′-disubstituted B rings. The other aromatic proton signals at δ 6.20 (1H, d, *J* = 2.10 Hz, H-6) and 6.40 (1H, d, *J* = 2.05 Hz, H-8) were apparently due to meta-coupled protons on the flavonoid’s A ring, while one methoxy group was evident at δ 3.78 (3H, s). Other signals in the ^13^C-NMR (CD_3_OD) were observed at δ 94.7 (C-8), 99.7 (C-6), 105.8 (C-10), 116.4 (C-2′), 116.4 (C-5′), 122.3 (C-6′), 122.9 (C-1′), 139.5 (C-3), 146.5 (C-3′), 150.0 (C-4′), 158.0 (C-9), 158.4 (C-2), 163.1 (C-5), 166.0 (C-7), and 180.0 (C-4). The carbon resonance at δ 60.5 was due to the presence of the methoxy group. These data agree with literature reports on 2-(3,4-dihydroxyphenyl)-5,7-dihydroxy-3-methoxy-4H-chromen-4-one [(3-O-methyl quercetin) (C_16_H_12_O_7_)] [15,16]. They were also confirmed by ^1^H-^13^C correlations in heteronuclear single quantum correlation spectroscopy (HSQC) measurement, which showed that three hydrogen atoms were directly attached to the carbon.

The alpha-glucosidase inhibition test of compounds **1** and **2** showed that compound **1** had a significant impact compared to compound **2**, with the compounds recording IC_50_ values of 6.53 ± 0.16 µg/mL and 39.95 ± 1.59 µg/mL, respectively (Table 3).

### 2.3. Luteolin Determination

TLC-densitometry was one of the methods used to measure active substance amounts. It was also used to determine the luteolin levels, with silica F254 being used as the stationary phase. Each sample was eluted thrice at the same TLC plate using *n*-hexane/ethyl acetate/formic acid (35:15:0.5). The area under curve (AUC) values were determined by scanning the dried spot with a TLC scanner at a 360 nm wavelength. A linear regression equation (y = 48.534*x* − 27,148, R² = 0.99) was derived by plotting the luteolin concentrations against the AUC values. The luteolin concentrations of 600, 700, 800, 900, and 1000 µg/mL produced AUC values of 2466.4, 6030.5, 12,330, 15,643, and 21,927 AU, respectively. These were used to calculate the luteolin levels in the crude drug, methanolic extract, and ethyl acetate fraction. In the triplicate test, the results showed that 20,000 µg/mL of the crude drug flower solution yielded values of 7481.1, 5810.5, and 4015.5, the methanolic flower extract yielded values of 3827.9, 5127.5, and 3793, and the acetate flower fraction yielded values of 4429.1, 7016.2, and 7456.4. Additionally, one gram of the crude drug, methanolic extract, and ethyl acetate fraction was found to contain 3.4 ± 0.2 mg (0.3%), 32.4 ± 0.8 mg (3.2%), and 68.9 ± 3.4 mg (6.9%) of luteolin, respectively.

## 3. Discussion

The alpha-glucosidase inhibitory activity of various parts of *G. amygdalinum* and its isolated compounds was investigated. A previous in vitro study indicated the leaves’ hypoglycemic effect. The MeOH extract of the *G. amygdalinum* leaf was reported to have significantly inhibited alpha-glucosidase [17]. Despite numerous reports on the secondary metabolite profile of the leaf, no alpha-glucosidase inhibition activity is yet observed in *G. amygdalinum* flower extracts and fractions. The activity of these plant extracts can be attributed to the phytoconstituents present in them. In this study, the methanolic flower extract of *G. amygdalinum* showed outstanding inhibitory alpha-glucosidase activity. It was further fractionated in different polarity solvents [18]. Among its fractions, the ethyl acetate fraction showed the maximum inhibitory activity. The ethyl acetate fraction is known to be rich in flavonoids. Flavonoids are phenolic substances with many health properties that frequently exhibit inhibitory effects against alpha-glucosidase enzymes [19]. In general, although previous studies reported on luteolin and its pharmacological activity [20,21,22], more analyses are performed on the leaves than the flower. Furthermore, four compounds that are reported to be isolated from flower extracts are tricosane, vernolide, isorhamnetin, and luteolin [20]. Most of the content in the ethyl acetate fraction is known as luteolin. This flavone is commonly found in Asteraceae [23]. It was found to contribute to the effect of alpha-glucosidase inhibitory activity [13], but to date, there are still no reports on luteolin in *G. amygdalinum*’s inhibitory effect against alpha-glucosidase. A previous report showed that luteolin has a non-competitive inhibition mechanism type [24]. Luteolin can also be found in *Vernonia cinera*, but its bioactivity is shown to be anti-inflammatory [12]. The OCH_3_ group presence in 3-O-methyl quercetin seems to be closely associated with these effects as well. Other plants that also contain luteolin are *Brassica oleracea*, *Capsicum annum*, *Capscium frustescens*, *Allium fistulosum*, *Averhoa bilimbi*, *Phaseolus vulgaris*, *Daucus carota*, *Raphanus sativus*, *Apium graveolens*, and *Garcinia atroviridus*. The luteolin content in those plants ranged between 9–1035.0 mg/kg; this fact explains why those edible plants have a lot of benefits for health maintenance [25].

## 4. Materials and Methods

### 4.1. Chemicals

Bovine serum albumin (EC Number 232-936-2, A8806), α-glucosidase (EC Number 3.2.1.20 Saccharomyces cerevisiae, G5003), and p-nitrophenyl-α-D-glucopyranoside (pNPG, N1377) were purchased from Sigma-Aldrich (St. Louis, MO, USA), while acarbose was purchased from TCI Chemicals (Hydrabad, India). A specific spray reagent, citroborate (5 g citric acid, 5 g boric acid, and 100 mL ethanol), and 10% (*v*/*v*) H_2_SO_4_ in a MeOH solution were used to identify flavonoids and other compounds. Additionally, during extraction and separation, the entire solvents were distilled before use, making them of technical grade.

### 4.2. Plant Materials

*G. amygdalinum* parts, namely roots, stem barks, leaves, and flowers, were collected on 30 October 2019, from Bandung, located in West Java, Indonesia. The plant specimens were identified and authenticated at the Herbarium Bogoriense of the Indonesian Institute of Sciences Research Center for Biology. The specimen number is B-139/2021.

### 4.3. Phytochemical Screening

The phytochemical screening was performed in methanolic flower extracts using the standard procedure [26].

### 4.4. Plant Extraction

The powder of *G. amygdalinum* dried roots (600 g), stem barks (600 g), leaves (1.6 kg), and flowers (400 g) were separated and extracted by maceration at room temperature in absolute MeOH (5 L × 3 times). Each crude extract was evaporated with a rotary evaporator, then filtered and concentrated to yield 52 g, 100 g, 273 g, and 80 g, respectively.

### 4.5. Inhibitory Assay of Alpha-Glucosidase

Alpha-glucosidase inhibition was assessed using the adopted method with slight modifications [27]. Phosphate buffer solution (36 μL), a sample solution with different concentrations (30 mL), and pNPG substrate with 6 mM concentration (17 μL) were successively added to 96 microplate wells. The incubation of this mixture was performed at 37 °C for 5 min. In each well, 17 μL of alpha-glucosidase solution at 0.2 U/mL was added to obtain a total volume of 100 mL, which was followed by incubation at 37 °C for 15 min. Furthermore, the initial reaction was brought to completion by the addition of Na_2_CO_3_ solution (100 μL) at 200 mM. At 400 nm, the entire absorbances were measured in triplicate with a Microplate Reader (Tecan^®^), while the positive control was utilized as acarbose. Percentage inhibition was used to express the inhibitory activity of alpha-glucosidase, while the calculation followed the formula below:

Inhibition percentage (%) = (B1–B2)/B1 × 100%.

B1 = Blank absorbance * − control of blank absorbance **

B2 = Sample absorbance − control of sample absorbance ***

* Blank contains PBS + pNPG + enzyme + Na_2_CO_3_

** Control of blank contains PBS + Na_2_CO_3_, without enzyme

*** Control of sample contains extract + PBS + Na_2_CO_3_, without enzyme

The IC50 was calculated using a linear regression equation (y = a + bx) in which the sample concentration was represented on the x-axis and the percentage inhibition was represented on the y-axis.

### 4.6. Isolation and Structure Determination of Active Compounds

The dried flowers of *G. amygdalinum* were extracted through maceration with MeOH (5 L × 3 times). The total extraction (80 g) of MeOH was suspended in 20% MeOH in 200 mL of water and partitioned sequentially with *n*-hexane and ethyl acetate, yielding *n*-Hexane (A 10g), ethyl acetate (B 25 g), and water (C 35 g). The ethyl acetate fraction (24 g) was subjected to silica gel vacuum column chromatography with a gradient of *n*-hexane (100%), *n*-hexane–ethyl acetate (9:1; 8:2; 7:3; 6:4; 5:5; 4:6; 3:7; 2:8; 1:9 for each step), and MeOH (100%) to produce 20 fractions. The compound profiles were analyzed using TLC under UV at λ 254 and 366 nm by spraying the compounds with citroborate. Fractions 5, 6, and 7 were combined and isolated with a gradient of *n*-hexane (100%), *n*-hexane–chloroform (9:1; 8:2; 7:3; 6:4; 5:5; 4:6; 3:7; 2:8; 1:9 for each step), and MeOH (100%) to produce 50 fractions (Fr. B1-50). Subfraction B40 was separated using silica gel column chromatography with an isocratic of *n*-hexane–ethyl acetate (4:1) to produce compound **1** and compound **2**.

### 4.7. Determination of Luteolin by TLC-Densitometric Method

About 3 µL of each calibration sample was spotted on TLC silica gel F_254_. Further, various luteolin concentrations were prepared for a calibration curve. Then, each stock solution of the crude drug (50 g in 50 mL MeOH), methanolic extract (100 mg in 5 mL MeOH), and ethyl acetate fraction (100 mg in 5 mL MeOH) was prepared, and the experiments were performed in triplicate. Furthermore, the TLC-densitometry by Camag TLC Scanner 3^®^ was used.

### 4.8. Statistical Analysis

The alpha-glucosidase inhibitory activity data of plant extracts and isolated compounds were presented as mean ± standard deviation (S.D.). The data were further analyzed using the independent-samples *t*-test or the Mann–Whitney U test and the one-way analysis of variances (ANOVA) using SPSS version 25.0 software. *p*-values less than 0.05 were indicated as significant.

## 5. Conclusions

The activity assays showed that the *G. amygdalinum* extracts, fractions, and isolated compounds exerted alpha-glucosidase inhibitory activities compared to the positive control, acarbose. The flower extract had a significant inhibitory effect, while the bioactive compounds included luteolin and 3-O-methyl quercetin. This is the first report on the AGI activity and 3-O-methyl quercetin isolation of the *G. amygdalinum* flower.

## Figures and Tables

**Figure 1 molecules-27-02132-f001:**
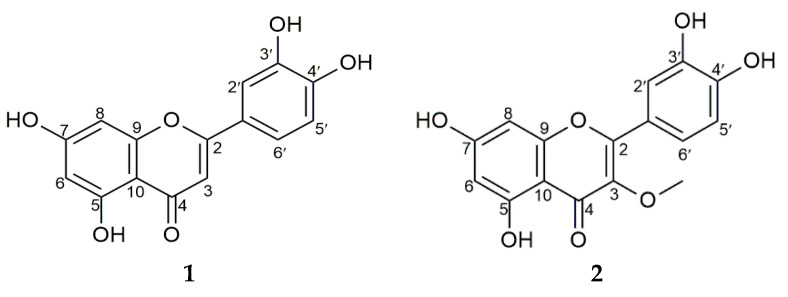
Structures of compounds **1** and **2**.

**Table 1 molecules-27-02132-t001:** The extracts’ alpha-glucosidase inhibitory activity assay.

Samples	Inhibition (%)	
	100 µg/mL	200 µg/mL
Root extract	4.66 ± 0.39 *	8.07 ± 0.70 *
Stem bark extract	3.63 ± 0.56 *	5.98 ± 0.60 *
Leaf extract	6.02 ± 1.38 *	8.18 ± 0.85 *
Flower extract	59.34 ± 1.26 *	73.57 ± 0.83 *
Acarbose	42.13 ± 0.12	66.03 ± 1.11

Note: Mean ± SD, *n* = 3. The symbol * represents a statistically significant difference from the acarbose of each concentration of *p* < 0.05, analyzed by the independent-samples *t*-test or Mann–Whitney U test.

**Table 2 molecules-27-02132-t002:** The flower fractions’ alpha-glucosidase inhibitory activity assay.

Samples	Inhibition (in %)	
	100 µg/mL	200 µg/mL
Water fraction	13.37 ± 1.12 *	22.95 ± 0.73 *
Ethyl acetate fraction	82.11 ± 4.20 *	87.63 ± 0.78 *
*n*-Hexane fraction	15.43 ± 0.44 *	32.59 ± 0.80 *
Acarbose	43.13 ± 0.92	66.83 ± 2.11

Note: Mean ± SD, *n* = 3. The symbol * represents a statistically significant difference from the acarbose of each concentration sample of *p* < 0.05, analyzed by the independent-samples *t*-test or Mann–Whitney U test.

**Table 3 molecules-27-02132-t003:** The IC_50_ values of extract, fraction, and isolated compound.

Samples	IC_50_ (µg/mL)
Methanolic flower extract Ethyl acetate fraction	47.29 ± 1.12 * 19.24 ± 0.12 *
Compound **1**	6.53 ± 0.16 *
Compound **2**	38.95 ± 1.59 *
Acarbose	73.36 ± 3.05

Note: Mean ± SD, *n* = 3. The symbol * represents a statistically significant difference compared with the acarbose, analyzed by the ANOVA test and mean Duncan’s multiple range test compared values; * *p* < 0.05.

## Data Availability

Not applicable.

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
