# Peer review of "In Vitro Alpha-Glucosidase Inhibitory Activity and the Isolation of Luteolin from the Flower of Gymnanthemum amygdalinum (Delile) Sch. Bip ex Walp."

_molecules, 2022, doi:10.3390/molecules27072132_

Round 1

Reviewer 1 Report

Luteolin often present as glycoside in plant.

This manuscript will be more interesting if the authors can add the contents of luteolin from ethyl acetate extracts from different plant parts as well. 

  1. Page 1, line 3-4: Gymnanthemum amygdalinum (Delile) Sch. Bip should be italicized and the author name for this plant should be checked from the latest nomenclatural database. Gymnanthemum amygdalinum (Delile) Sch. Bip ex Walp.
  2. Page 1, line 3: bioactive compounds can be changed to luteolin? Flower part also can be mentioned in the title.
  3. Page 1, line 13: Flowers extract should be changed to flower extract
  4. Page 1, line 24: Gymnanthemum amygdalinum; Vernonia amygdalina should be italicized
  5. Page1, line 34-35: “One of them is Gymnanthemum amygdalinum, also known as Vernonia amygdalina, which belongs to the Asteraceae family”. Please provide literature review on its traditional use to treat diabetes.
  6. Page 1: Must add more literature review in the introduction section on the compounds found in the plant. Also must add previous studies on luteolin as inhibitor of alpha glucosidase and luteolin in Asteraceae.
  7. Page 2, line 48: The occurrence of Table 2 should be mentioned here
  8. Page 2, Table 1: Leave should be spelled leaf
  9. Page 5, line 182: More discussion on luteolin is needed
  10. Page 6, line 257: This manuscript will be more interesting if the authors can add the contents of luteolin from different plant parts using ethyl acetate extracts as well in “4.7 Determination of Luteolin by TLC-Densitometric Method”
  11. Many grammatical errors though out the manuscript that should be improved

Author Response

Reviewer comments : 

  1. Page 1, line 3-4: Gymnanthemum amygdalinum (Delile) Sch.Bip should be italicized and the author name for this plantshould be checked from the latest nomenclatural database.Gymnanthemum amygdalinum (Delile) Sch. Bip ex Walp. Response 1 : Species name was italicized.

2. Page 1, line 3: bioactive compounds can be changed toluteolin? Flower part also can be mentioned in the title. Response 2 : bioactive was changed to luteolin, and the flower part was mentioned in the title.

3. Page 1, line 13: Flowers extract should be changed to flowerextract. Response 3 : The flower extract is changed to flowerextract.

4. Page 1, line 24: Gymnanthemum amygdalinum; Vernoniaamygdalina should be italicized. Response 4 : The name of species is now italicized.

5. Page1, line 34-35: “One of them is Gymnanthemumamygdalinum, also known as Vernonia amygdalina, whichbelongs to the Asteraceae family”. Please provide literaturereview on its traditional use to treat diabetes. Response 5 : Literature review wass added to Page 1 line 34-35.

6. Page 1: Must add more literature review in the introductionsection on the compounds found in the plant. Also must addprevious studies on luteolin as inhibitor of alpha glucosidaseand luteolin in Asteraceae. Response 6 : Previous studies of luteolin were added.

7. Page 2, line 48: The occurrence of Table 2 should bementioned here. Response 7 : Table 2 was mentioned.

8. Page 2, Table 1: Leave should be spelled leaf. Response 8 : Leave was changed to leaf.

9. Page 5, line 182: More discussion on luteolin is needed. Response 9 : more discussion about luteolin was added.

10. Page 6, line 257: This manuscript will be more interesting if theauthors can add the contents of luteolin from different plantparts using ethyl acetate extracts as well in “4.7 Determinationof Luteolin by TLC-Densitometric Method” . Response 10 : The contents of luteolin were added.

11. Many grammatical errors though out the manuscript that shouldbe improved. Response 11 : The manuscript will be sent to MDPI English editing pararelly.

Reviewer 2 Report

The manuscript described the bioactive-guided isolation of Gymnanthemum amygdalinum following the alpha-glucosidase inhibition. Although the data of alpha-glucosidase inhibition of this plant are scarce, two isolated compounds are well-known. The scientific contents of the current version are not sufficient for Molecules readers. The authors were suggested to investigate the kinetic mechanism of compound 2 or the in silico study if possible in order to improve the scientific data. Alpha-glucosidase inhibition of luteolin has been studied deeply, thus, please make a literature review in the manuscript regarding this compound.

Other points:

  • The phytochemical screening is provided but there is no data about this.
  • Lacking the voucher specimen of the studied plant.
  • Revised L64. How were the chemical structures visualized by UV and NMR?
  • Please rewrite the isolation part following the newest papers of Molecules. The current content is complicated to read.

Author Response

The manuscript described the bioactive-guided isolation of Gymnanthemum amygdalinum following the alpha-glucosidase inhibition. Although the data of alpha-glucosidase inhibition of this plant are scarce, two isolated compounds are well-known. The scientific contents of the current version are not sufficient for Molecules readers. The authors were suggested to investigate the kinetic mechanism of compound 2 or the in silico study if possible in order to improve the scientific data. Alpha-glucosidase inhibition of luteolin has been studied deeply, thus, please make a literature review in the manuscript regarding this compound.

Rensponse : The previous studies about the kinetic mechanism and in silico study were added to the introduction for the literature review.

Other points:

  • The phytochemical screening is provided but there is no data about this. Response : The phytochemical screening results were added.
  • Lacking the voucher specimen of the studied plant. Response : The voucher specimen was added
  • Revised L64. How were the chemical structures visualized by UV and NMR? Response : The methods for chemical structures visualized by UV and NMR were added.
  • Please rewrite the isolation part following the newest papers of Molecules. The current content is complicated to read. Response : The isolation part was rewritten.

Round 2

Reviewer 1 Report

  1. Page 1, line 11: “its roots, stem barks, leaves, and flower extracts”. Why singular and plural were mixed here?
  2. Table 3: Still typo “Mehtanolic flower extract”
  3. Page 10, line 413-418: Should change the title to lowercase.

25) Panda SK, Luyten W. Antiparasitic Activity in Asteraceae with Special 413 Attention to ethnobotanical use by the tribes of Odisha India, Parasite, 25, 1- 414 24, (2018). 415

26) Djeujo FM, Raggazi E, Urettini M, Sauro B, Chicero E, Tonelli M, Froldi G. 416 Magnolol and Luteolin Inhibition of a-Glucosidase activity : Kinetics and 417 Type of Interaction Detected by in vitro and in silico studies, Pharmaceuticals, 418 15, 1-22.

Comment10 in version 1

Page 6, line 257: This manuscript will be more interesting if theauthors can add the contents of luteolin from different plant parts using ethyl acetate extracts as well in “4.7 Determination of Luteolin by TLC-Densitometric Method” . Response 10 : The contents of luteolin were added.

The contents of luteolin in different plant parts were not added.

Author Response

Page 1, line 11: “its roots, stem barks, leaves, and flower extracts”. Why singular and plural were mixed here? corrected

Table 3: Still typo “Mehtanolic flower extract” corrected

Page 10, line 413-418: Should change the title to lowercase. corrected

25) Panda SK, Luyten W. Antiparasitic Activity in Asteraceae with Special 413 Attention to ethnobotanical use by the tribes of Odisha India, Parasite, 25, 1- 414 24, (2018). corrected

415 26) Djeujo FM, Raggazi E, Urettini M, Sauro B, Chicero E, Tonelli M, Froldi G. 416 Magnolol and Luteolin Inhibition of a-Glucosidase activity : Kinetics corrected

and 417 Type of Interaction Detected by in vitro and in silico studies, Pharmaceuticals, 418 15, 1-22. corrected

Comment10 in version 1 Page 6, line 257: This manuscript will be more interesting if theauthors can add the contents of luteolin from different plant parts using ethyl acetate extracts as well in “4.7 Determination of Luteolin by TLC-Densitometric Method” . Response 10 : The contents of luteolin were added. The contents of luteolin in different plant parts were not added. (done) it is now in page 5 line 468-473.

Reviewer 2 Report

The new version of the manuscript has been revised thoroughly. 

Author Response

Our manuscript was sent to the MDPI English editor. Herewith I send the certificate.
